# Fungal Infection Testing in Pediatric Intensive Care Units—A Single Center Experience

**DOI:** 10.3390/ijerph19031716

**Published:** 2022-02-02

**Authors:** Joanna Klepacka, Zuzanna Zakrzewska, Małgorzata Czogała, Magdalena Wojtaszek-Główka, Emil Krzysztofik, Wojciech Czogała, Szymon Skoczeń

**Affiliations:** 1Department of Microbiology, University Children’s Hospital, Wielicka 265, 30-663 Krakow, Poland; jklepacka@usdk.pl (J.K.); czarnohora@interia.pl (W.C.); 2Department of Oncology and Hematology, University Children’s Hospital, Wielicka 265, 30-663 Krakow, Poland; zuzanna.zakrzewska@doctoral.uj.edu.pl (Z.Z.); malgorzata.czogala@uj.edu.pl (M.C.); 3Department of Pediatric Oncology and Hematology, Institute of Pediatrics, Jagiellonian University Medical College, 30-663 Krakow, Poland; 4Student Scientific Group of Pediatric Oncology and Hematology, Jagiellonian University Medical College, 30-663 Krakow, Poland; magda.wojtaszek-glowka@student.uj.edu.pl (M.W.-G.); emilkrzysztofik.ek@gmail.com (E.K.)

**Keywords:** mycological diagnostic, fungal infections, intensive care unit

## Abstract

Mycoses are diseases caused by fungi that involve different parts of the body and can generate dangerous treatment complications. This study aims to analyze fungal infection epidemiology in intensive care units (Pediatric and Cardiac Surgery Intensive Care Units—PCICU) and the Neonatal Intensive Care Unit (NICU) in one large pediatric center in the period 2015–2020 compared with 2005. The year 2005 was randomly selected as a historical time reference to notice possible changes. In 2005 and 2015–2020, 23,334 mycological tests were performed in intensive care units. A total of 4628 tests (19.8%) were performed in the intensive care units. Microbiological diagnostics involved mycological and serological testing. Of the 458 children hospitalized in the NICU, positive results in the mycological tests in the studied years were found in 21–27% of the children and out of 1056 PCICU patients, positive results were noticed in 18–29%. In both departments, the main detected pathogen was *Candida albicans* which is comparable with data published in other centers. Our experience indicates that blood cultures as well as the detection of antifungal antibodies do not add important information to mycological diagnostics. For the years of observation, only a few positive results were detected, even in patients with invasive fungal diseases. To our knowledge, this is one of a few similar studies over recent years and it provides contemporary reports of mycoses in pediatric ICU patients.

## 1. Introduction

Mycoses or fungal diseases can be divided into superficial, subcutaneous, and systemic. Invasive fungal infection (IFI) is a severe, systemic, life-threatening disease caused by fungi [1]. IFIs are a leading cause of morbidity and mortality in immunocompromised children and neonates [2]. However, even the common fungal infection could be dangerous for a severely ill child as it leads to an inflammatory reaction, an additional therapy risk, or a delay in treatment.

The majority of our knowledge comes from adult population or pediatric cancer patient single center studies [3,4]. Nonspecific symptoms, diagnostic challenges, and the occasional usage of anti-fungal prophylaxis could be the reasons why many cases of fungal infections remain undiagnosed. According to the revised criteria of the EORTC/MSG Consensus Group, adequate microbiological evaluation is the key to the accurate diagnosis of fungal infections ensuring appropriate treatment [5].

The study was conducted in a tertiary care pediatric referral center in southern Poland that is comprised of 24 departments including 3 pediatric intensive care units and a separate neonatal intensive care unit. The hospital has 33,000 admissions per year and performs approximately 7000 operations including 450 cardiac surgery procedures per year on average. In the intensive care units, children with the most advanced forms of various congenital defects and burns, or neonates with extremely low birth weights are managed. Moreover, intensive therapy departments provide care for patients after complex surgeries (including cardiac and brain procedures), after accidents, or with severe complications after multimodal oncological treatment. Therefore, children treated in these units are at a high risk of hospital-acquired infections as some of them are immunocompromised or malnourished, or are undergoing combined antibiotic therapy which makes them a population extremely prone to fungal infections.

The hospital provides comprehensive diagnostic services for every patient with a suspected fungal infection. Various radiological imaging methods help patients with advanced mycoses to be recognized. Surgical procedures are often necessary to obtain the appropriate material. In the Microbiology Department, various diagnostic methods are routinely ordered including fungal cultures on samples from several body regions. Moreover, detailed serological tests are used to expand the patient’s evaluation.

The aim of the study was to analyze the epidemiology of fungal infections in the intensive care units (Pediatric and Cardiac Surgery Intensive Care Units—PCICU) and the Neonatal Intensive Care Unit (NICU) of the Children’s University Hospital (UCH) in Krakow in the period 2015–2020. The data were compared with previous data collected in 2005 to notice possible changes.

## 2. Materials and Methods

In 2005 and 2015–2020, the Clinical Microbiology Department of UCH performed a total of 23,334 mycological tests. A total of 4628 tests (19.8%) were performed in the intensive care units, including 3755 tests for (PCICU) and 873 for NICU. The year 2005 was used as a baseline to estimate changes in the amount of performed tests and cultured fungi.

Microbiological diagnostics involved mycological and serological testing.

Analyzed samples were divided into 5 groups based on the source: (1) gastrointestinal tract (stool, anal swabs), (2) upper respiratory tract (nasopharyngeal, nasal, and throat swabs), (3) lower respiratory tract (bronchial lavage, bronchoalveolar lavage, intrapleural fluid, aspirates), (4) urinary tract (urine), (5) other (collected from skin lesions, urethral catheter, vaginal smears, eye swabs, wound swabs).

All children admitted to the intensive care units who had a mycological test performed were included. The first group hospitalized in PCICU were composed of 1056 patients (595 males). The numbers in the studied years were as follows: 2005—102, 2015—147, 2016—163, 2017—219, 2018—189, 2019—138, 2020—98. In 297 patients (163 males) serological mycological tests were performed: 2015—71, 2016—58, 2017—47, 2018—35, 2019—50, 2020—36. Detailed information can be found in Table 1.

The NICU group was composed of 458 patients (232 males) who had mycological tests. The numbers in the studied years were as follows: 2005—64, 2015—76, 2016—65, 2017—70, 2018—51, 2019—68, 2020—64. A group of 11 patients (4 males) had serological mycological tests: 2015—4, 2016—1, 2017—1, 2018—1, 2019—2, 2020—2. Detailed information is presented in Table 2.

Mycological testing included:Direct and stained samples;Cultures on liquid and/or Sabourauda agar medium with/without antibiotics at a temperature of 30 °C;Identification of cultivated yeasts;
a.Filamentation (germ tube test); b.Differential surface culture (corn meal agar, differentiating presence or lack of chlamydospores and/or pseudomycelium;c.Pre-prepared biochemical tests ID 32 C (bioMerieux), BD Phoenix Yeast (Bacton Dickinson);d.Mass spectometry (MALDI-TOF MS) using matrix-assisted desorption/laser ionisation time-of-flight;

Serological diagnostics of fungal infections consisted of:Anty-mannan Candida antibodies in serum or plasma (PLATELIA Candida Ab Plus, BIO-RAD);IgG anti-Aspergillus antibodies in serum or plasma (PLATELIA Aspergillus IgG, BIO-RAD);Candida mannan antigen in serum or plasma (PLATELIA Candida Ag Plus, BIO-RAD);Galactomannan Aspergillus antigen in serum or plasma (PLATELIA Aspergillus Ag, BIO-RAD).

For the statistical analysis of the collected data, we used Statistica 13 (StatSoft, INC., Tulsa, OK, USA). The incidence of positive results for each year was compared using Pearson’s chi-squared test.

## 3. Results

During the chosen period, 563 out of 3755 mycological tests from PCICU were positive. Figure 1 shows the number of collected samples each year divided into five source groups: (1) gastrointestinal track—386 tests, (2) upper respiratory tract—27 tests, (3) lower respiratory tract—1356 tests, (4) urinary tract—587 tests, and (5) other—305 tests.

The distribution of the results collected from PCICU in the studied years, including the number of positive tests is shown in Figure 2.

During the studied period, 205 out of 873 mycological tests collected from the NICU were positive. Figure 3 shows the number of collected samples each year divided into five source groups: (1) gastrointestinal track—332 tests, (2) upper respiratory tract—15 tests, (3) lower respiratory tract—193 tests, (4) urinary tract—238 tests, and (5) other—65 tests.

The distribution of the results collected in the NICU in the studied years including the number of positive tests is shown in Figure 4.

Positive results from the mycological tests in PCICU were found in 41 out of 102 patients (40%) in 2005, 50/147 (34%) in 2015, 53/163 (32%) in 2016, 69/219 (31.5%) in 2017, 58/189 (30.6%) in 2018, 42/138 (30%) in 2019, and 27/98 (27%) in 2020, see Figure 5. The differences were not statistically significant (*p* > 0.05).

Positive results in NICU patients were found in 21 out of 64 patients (32.8%) in 2005, 21/76 (27.6%) in 2015, 21/65 (32%) in 2016, 18/70 (25.7%) in 2017, 17/51 (33%) in 2018, 16/68 (23.5%) in 2019, and 14/64 (21.8%) in 2020, see Figure 6. The differences were not statistically significant.

*Candida albicans* was the most frequently isolated yeast species each year from the samples collected in PCICU and NICU.

The prevalence of positive results in PCICU in the studied years was as follows: 2005—78% (42/54), 2015—53% (58/110), 2016—51% (48/94), 2017—44% (51/115), 2018—51% (43/85), 2019—59% (36/61), 2020—55% (24/44). The remaining species were isolated with various frequencies each year (0–289) and did not exceed 20% of all cultures. The details are shown in Figure 7.

The prevalence of positive *Candida parapsilosis* in the studied years was as follows: 2005—9% (5/54), 2015—4.5% (5/110), 2016—8.5% (8/94), 2017—19% (22/115), 2018—9.4% (8/85), 2019—9.8% (6/61), 2020—18% (8/44).

A significant increase in the positive cultures of *Candida dubliniensis* was observed: 2005—0% (0/54), 2015—5.5% (6/110), 2016—6.3% (6/94), 2017—7% (8/115), 2018—19% (16/85), 2019—15% (9/61), 2020—9% (4/44).

The same was noticed for *Candida glabrata*: 2005—9% (5/54), 2015—18% (20/110), 2016—14% (13/94), 2017—12% (14/115), 2018—7% (6/85), 2019—6.5% (4/61), 2020—9% (4/44), as well as *Candida lusitaniae*: 2005—0% (0/54), 2015—4.5% (5/110), 2016—8.5% (8/94), 2017—4% (5/115), 2018—7% (6/85), 2019—5% (3/61), 2020—13.6% (6/44) and *Candida krusei* was observed: 2005—0% (0/54), 2015—7% (8/110), 2016‚4% (4/94), 2017—5% (6/115), 2018—4.7% (4/85), 2019—6.5% (4/61), 2020—6.8% (3/44).

A significant increase in the positive cultures of *Saccharomyces spp.* was also observed: 2005—1.8% (1/54), 2015—1.8% (2/110), 2016—4% (4/94), 2017—5% (6/115), 2018—3.5% (3/85), 2019—3.3% (2/61), 2020—0% (0/44).

The percentages of positive isolate results in NICU patients were as follows: 2005—79% (23/29), 2015—72% (26/36), 2016—71% (22/31), 2017—58% (18/31), 2018—67% (18/27), 2019—67% (16/24), 2020—67% (18/27). The remaining species were isolated with various frequencies each year (0–53) and did not exceed 10% of all cultures. The details are shown in Figure 8.

The analysis of the samples collected from 244 PCICU patients from the gastrointestinal tract revealed positive results as follows: 2005—61% (17/28), 2015—40% (28/70), 2016—41% (26/64), 2017—24% (12/51), 2018—41% (13/32), 2019—43% (12/28), and 2020—41% (9/22). A statistically significant difference between the positive and negative results was found in a comparison between 2005 and 2017 (*p* = 0.0010). The most frequently isolated species in the gastrointestinal tract samples can be found in Figure 9.

The analysis of the samples collected from 260 NICU patients from the gastrointestinal tract revealed positive results as follows: 2005—35% (13/37), 2015—30% (14/46), 2016—36% (12/33), 2017—47% (14/30), 2018—35% (9/26), 2019—33% (14/42), and 2020—26% (12/46).

No statistically significant differences between the positive and negative results were found. The most frequently isolated species in the gastrointestinal tract NICU samples can be found in Figure 10.

The analysis of the samples collected from 25 PCICU patients (27 probes in total) from the upper respiratory tract revealed a total of 14 yeast isolates. More details are shown in Table 3.

The prevalence of positive results in the studied years was as follows: 2005—33% (2/6), 2015—67% (2/3), 2016—20% (1/5), 2017—75% (3/4), 2018—100% (1/1), 2019—40% (2/5), and 2020—100% (1/1). No statistically significant differences between the positive and negative results were found.

Upper respiratory tract samples from NICU patients were collected from 14 patients (15 probes in total). Four yeast isolates were found. More details are presented in Table 4. The prevalence of positive results in the studied years was as follows: 2005—0%, 2015—100% (2/2), 2016—17% (1/6), 2017—0%, 2018—100% (1/1), 2019—0%, and 2020—0%. No statistically significant differences between the positive and negative results were found.

Lower respiratory tract samples from PCICU patients were collected from 668 patients (1396 probes in total). A total of 259 yeast isolates were found. More details are shown in Table 5.

The prevalence of positive results in the studied years was as follows: 2005—33% (15/46), 2015—32% (19/59), 2016—30% (30/101), 2017—30% (46/159), 2018—25% (36/146), 2019—28% (27/97), 2020—25% (15/60), with no statistically significant differences between the results.

Lower respiratory tract samples from NICU patients were collected in 141 patients (193 probes in total). A total of 27 yeast isolates were found. More details are shown in Table 6.

The prevalence of positive results in the studied years was as follows: 2005—0%, 2015—7% (1/15), 2016—21% (5/24), 2017—8% (3/34), 2018—15% (2/13), 2019—12% (3/24), 2020—22% (5/23), with no statistically significant differences between the results.

Urinary tract samples were collected in the group of 365 PCICU patients (587 probes in total) with 113 yeast isolates detected. More details can be found in Table 7. The prevalence of positive results in the studied years was as follows: 2005—26% (10/38), 2015—20% (12/60), 2016—11% (6/54), 2017—19% (11/57), 2018—23% (12/52), 2019—19% (9/47), 2020—14% (8/57), with no statistically significant differences between the results.

Urinary tract samples were collected in the group of 167 PCICU patients (238 probes in total) with 42 yeast isolates detected. More details can be found in Table 8. The prevalence of positive results in the studied years was as follows: 2005—28% (11/40), 2015—20% (7/35), 2016—12% (2/17), 2017—16% (3/19), 2018—38% (6/16), 2019—6% (1/16), 2020—7% (1/15), with statistically significant differences between the positive and negative results found in a comparison of 2018 and 2019 (*p* = 0.0415), as well as 2018 and 2020 (*p* = 0.0500).

The results of the study of antibody levels and the presence of positive fungal antigens (*Aspergillus* and *Candida*) in PCICU patients can be found in Table 9, Table 10, Table 11 and Table 12. The percentage of patients with the positive galactomannan Candida antigen was significantly higher in 2015 compared to 2017 (*p* = 0.0415) and 2020 (*p* = 0.0232) as well as in 2019 compared to 2017 (*p* = 0.0387) and 2020 (*p* = 0.0227).

The results of the study of antibody levels and the presence of positive fungal antigens (*Aspergillus* and *Candida*) in NICU patients can be found in Table 13, Table 14, Table 15 and Table 16.

## 4. Discussion

The study group was comprised of two subgroups of children treated in Intensive Care Units. Pediatric and Cardio Surgery Intensive Care Units (PCICU) patients are referred to the units due to complications from other hospital departments and after cardio surgical procedures. The primary reason for infections was severe conditions due to cardiac and respiratory disorders, multiorgan dysfunction, prolonged immobilization, secondary immunodeficiency, or the repeated cannulation of vessels. The Neonatal Intensive Care Unit (NICU) group is composed of premature babies and newborns with an increased risk of infections due to humoral immunodeficiency (decreased IgA, IgG, and IgM), decreased chemotaxis, low bone marrow reserve, decreased bowel movements, low acid concentration in the stomach, increased skin pH and permeability, as well as a lack of a proper composition of the microbiome [7]. The secondary factors that predispose patients to an increased rate of infections in those treated in all intensive care units were exposition to highly resistant pathogens due to colonization with hospital environment pathogens as well as the use of invasive diagnostics procedures and others such as mechanical ventilation, cannulation of large vessels, catheterization of the pulmonary artery and the urinary bladder, and intravenous alimentation. Together with prolonged hospitalization and repeated wide spectrum antibiotic therapy, the risk of fungal infection increases. In the 458 children presented (232 boys and 226 girls) that were hospitalized in NICU, positive results from the mycological tests in the studied years were found in 21–27% of the children. The main detected pathogen was *Candida albicans* which is comparable with data published in other centers [6,7,8,9]. Due to the common use of fluconazole in empiric antifungal prophylaxis in high-risk premature infants, an increased colonization of the gastrointestinal tract with Candida non-albicans was observed. The main pathogens were *Candida parapsilosis* (increase in all the studied years except 2017) and *Candida glabrata* in 2019 and 2020, which is in accordance with data published in the literature [8,9,10,11,12,13].

In 1056 PCICU patients (595 boys and 369 girls), positive results were noticed in 18–29%. The prevalence of patients with positive results gradually decreased in subsequent years: 2005—40%, 2015—34%, 2016—32%, 2017—31.5%, 2018—30.6%, 2019—30%, and 2020—27%. The decline was caused by the introduction of prophylaxis with fluconazole as a routine practice. Again, the most common pathogen was *Candida albicans* which is also found in other countries [6,8]. Escalation of the colonization of the gastrointestinal tract and an increased rate of infections caused by Candida non-albicans (mainly *Candida*
*krusei, Candida glabrata*) was observed. Similar data have been published in other centers [8,9,10]. An important difference from the published data concerned *Candida parapsilosis,* as we did not notice an increase in this species in isolates in subsequent (9%; 4.5%; 8.5%; 19%; 9.4%) years except 2017 and 2020 (9.8%; 18%). The common use of amphotericin B caused an elevation of *Candida lusitaniae* in the isolates in the years 2005 and 2015 (20%) compared to other years: 2005—0%, 2015—4.5%, 2016—8.5%, 2017—4%, 2018—7%, 2019—5%, 2020—13.6%.

Patients in intensive care units are treated according to numerous therapeutic protocols (with the utilization of invasive procedures such as intubation or tracheostomy), increasing the risk of hospital acquired pneumonia (HAP) and ventilation-acquired pneumonia (VAP). Aspiration of fluid from the gastrointestinal tract promoted by a linear position, a catheter in the stomach, or common duodenal–intestinal reflux ends with aspiration pneumonia. In PCICU patients, we observed such a relationship from studying the fungal species isolated from the lower respiratory tract in subsequent years: *Candida albicans* 17 (14.9%), 19 (14.1%), 19(8.6%), 31 (9.9%), 21(7%), 21 (13.4%), 11(9.4%); *Candida dubliniensis* 0%, 5 (3.7%), 5 (2.3%), 7 (2.2%), 12 (4%), 8 (5%), 1 (0.9%); *Candida parapsilosis* 3 (2.6%), 2 (1.5%), 4 (1.8%), 12 (3.8%), 4 (1.3%), 3 (2.6%); *Candida lusitaniae* 0%, 3 (2,2%), 5 (2,3%), 3 (1%), 5 (1.7%), 1 (0.6%), 2 (1.7%); *Candida glabrata* 0%, 3 (2,2%), 5 (2,3%), 3 (1%), 2 (0.7%), 1 (0.6%), 0%; *Candida krusei* 0%, 0%, 0%, 2 (0.6%), 1 (0.3%), 3 (1.9%), 1 (0.9%). Those species were typically isolated from the gastrointestinal tract and were aspirated to the airways.

Our experience indicates that blood cultures do not add important information to mycological diagnostics. For the years of observation, only a few positive results were detected even in patients with invasive fungal diseases.

The same was true in relation to the detection of antifungal antibodies. Our results showed that such detection was very rare, even though those patients were not immunocompromised. If detected, the increased level of antibodies could suggest the status of fungal infection. In all units in the analyzed period, we detected anti-Aspergillus antibodies in only one patient and anti-Candida in four patients. Detection of the fungal antigen can be more informative. In our patients, such detection was more common compared to antibodies. It should be treated as a screening, as the diagnostics of fungal infection are very difficult. It would be useful to establish validated methods of detection for fungal antigens in urine, fluid from respiratory airways, and stools. Unfortunately, colonization with fungal pathogens, especially *Candida albicans* is common, which makes the detection of its antigen less useful, but the methodology of the testing should be adapted to that.

The majority of the studies on fungal infections have concentrated on oncology departments. The problem of mycoses in intensive care units is underdiagnosed. This study showed the necessity of screening for fungal diseases in these high-risk patients. Moreover, we concluded that the diagnostic methods are limited. There is an urgent need to introduce new validated mycological tests in order to improve the survival of children with the most advanced medical conditions. The International Pediatric Fungal Network ((PFN) www.ipfn.org accessed on 21 October 2021) was created to facilitate international cooperation in terms of the understanding and management of pediatric fungal infections.

Our study has some limitations characteristic of most retrospective data analyses. We focused on microbiological tests results and did not include the clinical characteristics of the patients. The data came from only one large center and therefore the results could reflect patterns specific for this center. However, the results from multicenter studies that we referred to are comparable to ours. Moreover, we avoided selection bias as we enrolled all patients treated in the ICUs with suspicion of a fungal infection in the study. To our knowledge, only a few similar studies have been published recently and our study provides a contemporary report of mycoses in pediatric intensive care patients.

## 5. Conclusions

Our experience, based on our center, indicates that the percentage of positive mycological tests in pediatric and neonatal intensive care units reached 18–29%. In both departments, the main detected pathogen was *Candida albicans* which is comparable with data published in other centers [14,15,16,17]. Our experience indicates that blood cultures as well as the detection of antifungal antibodies do not add important information to mycological diagnostics. For the years of observation, only a few positive results were detected, even in patients with invasive fungal diseases.

## Figures and Tables

**Figure 1 ijerph-19-01716-f001:**
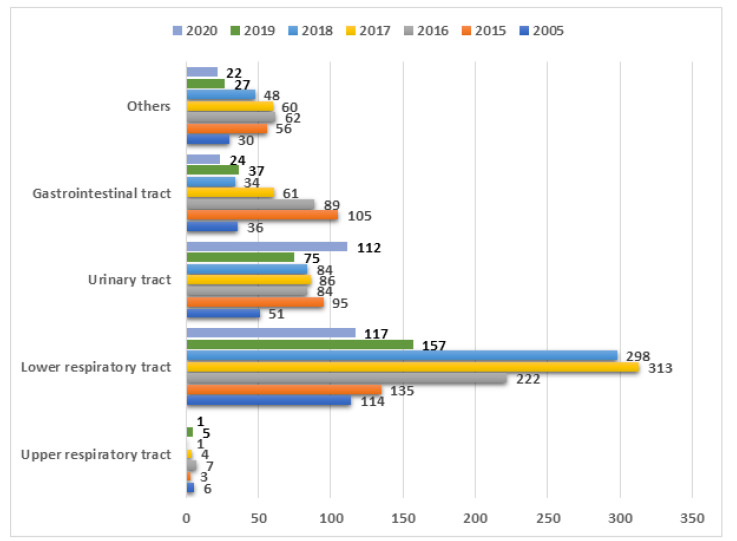
Number of samples from PCICU divided into 5 source groups.

**Figure 2 ijerph-19-01716-f002:**
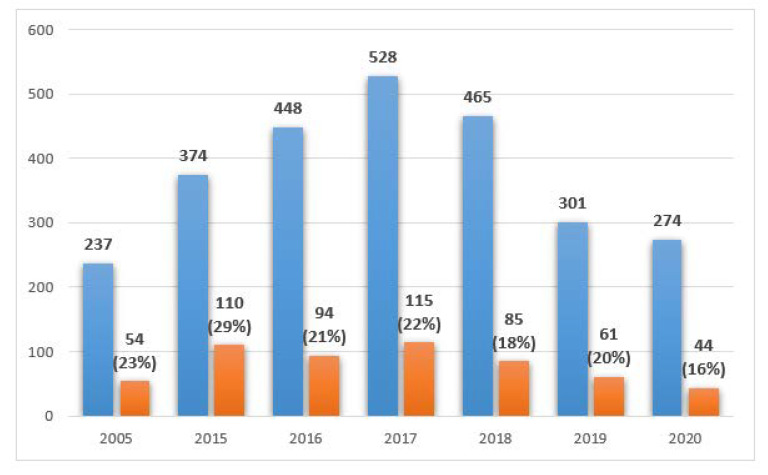
Mycological tests performed in PCICU in the studied years, including positive results.

**Figure 3 ijerph-19-01716-f003:**
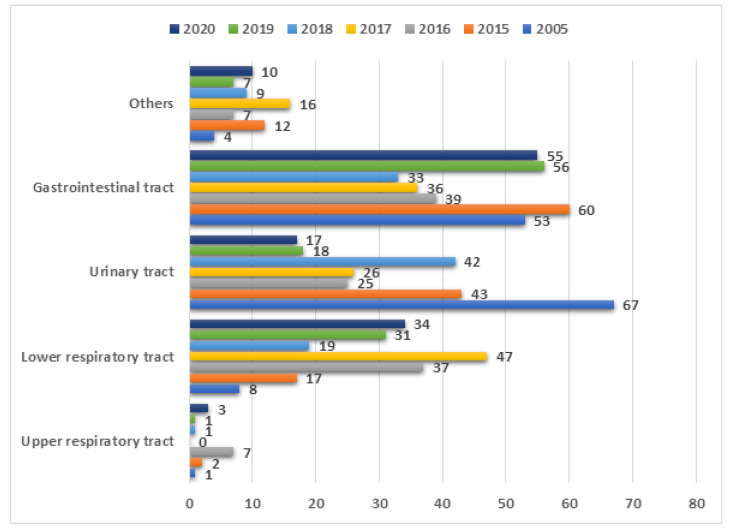
Samples collected in the NICU divided into 5 source groups.

**Figure 4 ijerph-19-01716-f004:**
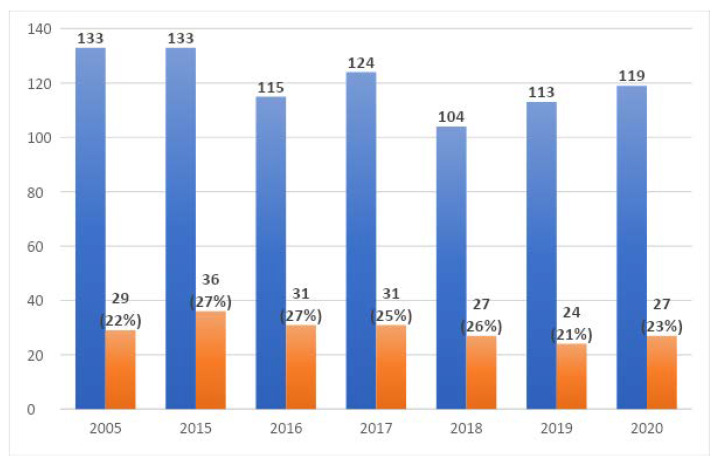
Mycological tests performed in the NICU in the studied years, including positive results.

**Figure 5 ijerph-19-01716-f005:**
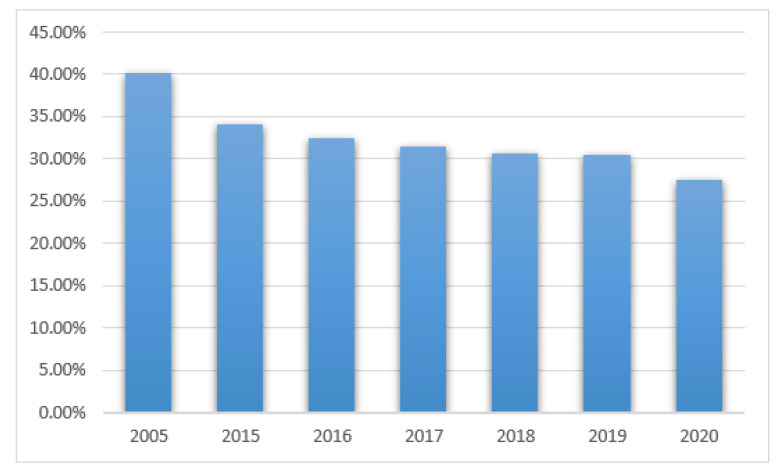
Positive results of mycological tests in PCICU patients in the studied years.

**Figure 6 ijerph-19-01716-f006:**
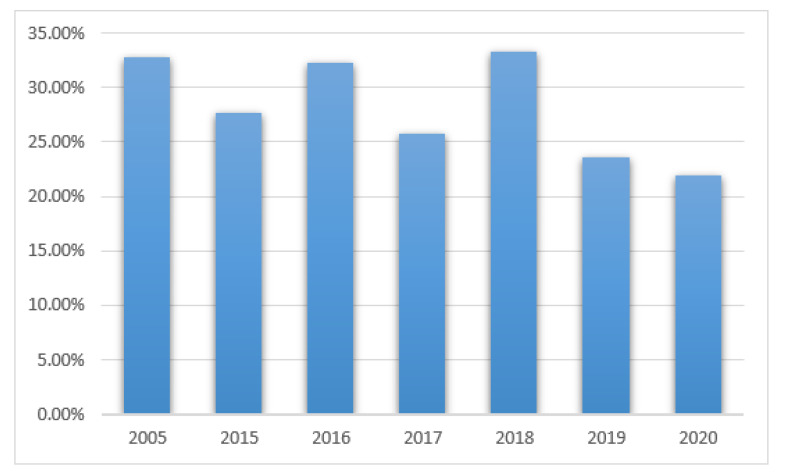
Positive results of mycological tests in NICU patients in the studied years.

**Figure 7 ijerph-19-01716-f007:**
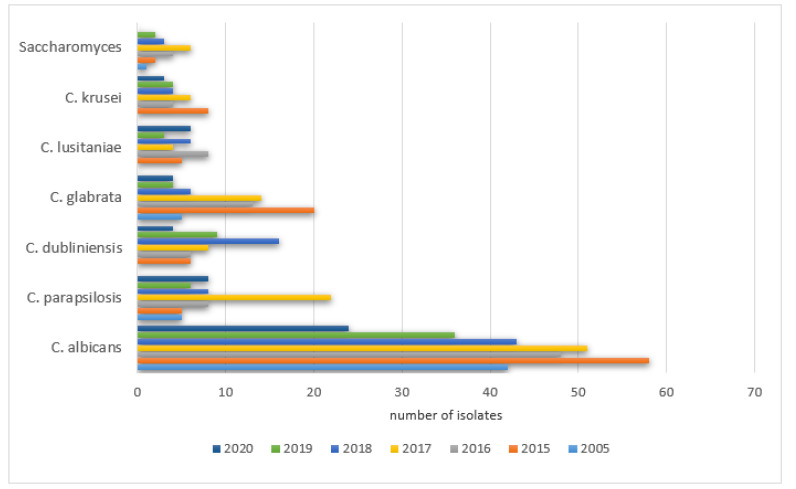
Most frequently isolated yeast species from samples collected in PCICU each year.

**Figure 8 ijerph-19-01716-f008:**
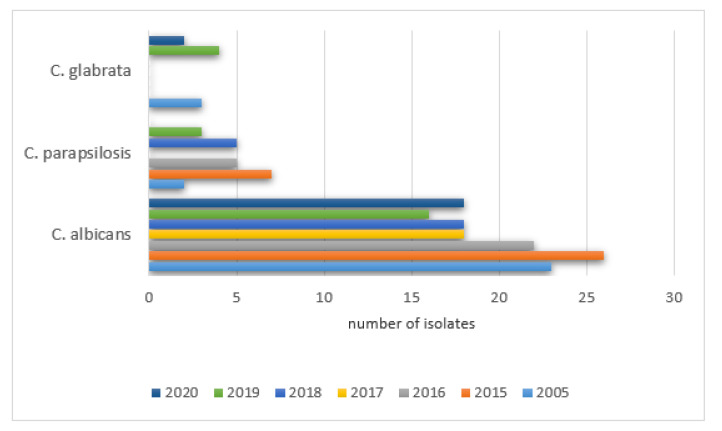
Most frequently isolated yeast species from samples collected in the NICU each year.

**Figure 9 ijerph-19-01716-f009:**
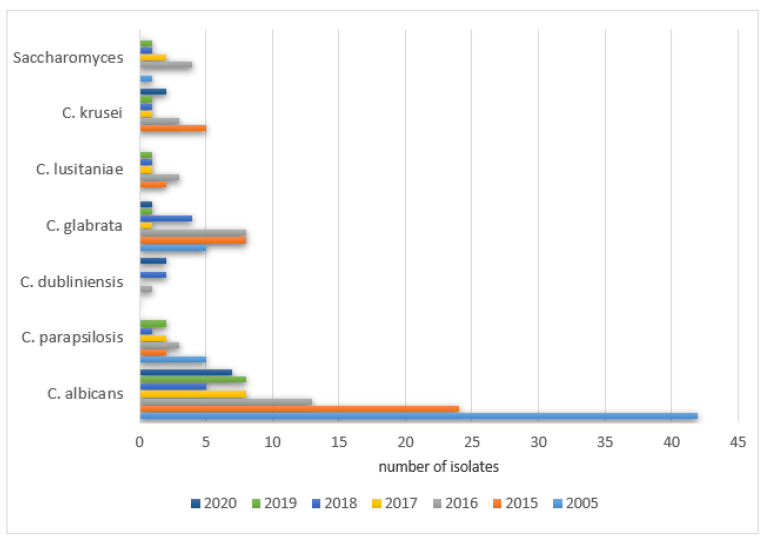
Yeast and *Saccharomyces* species isolated from gastrointestinal tract PCICU samples in the studied years.

**Figure 10 ijerph-19-01716-f010:**
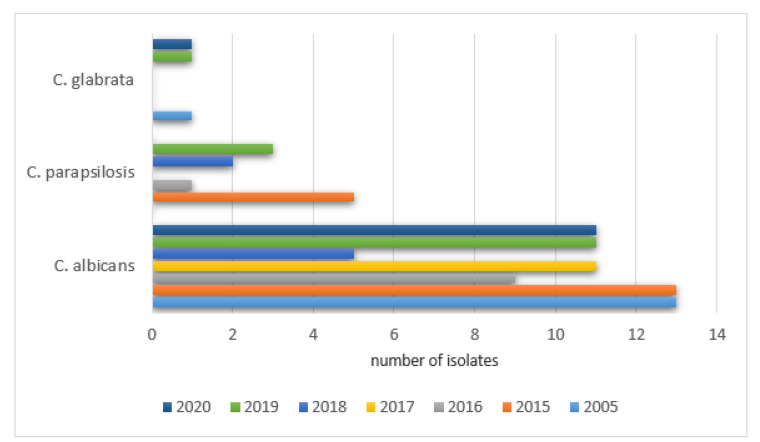
Yeast species isolated from gastrointestinal tract NICU samples in the studied years.

**Table 1 ijerph-19-01716-t001:** Mycological tests in PCICU patients.

	2005	2015	2016	2017	2018	2019	2020
Number of patients (with fungal cultures)	102	147	163	219	189	138	98
Female	48	60	65	90	87	69	42
Male	54	87	98	129	102	69	56
Mycological tests	237	374	448	528	465	301	274
Number of patients (with mycologic serological tests)	0	71	58	47	35	50	36
Female	0	34	21	20	20	25	14
Male	0	37	37	27	15	25	22
Serological tests	0	252	206	130	110	203	103

**Table 2 ijerph-19-01716-t002:** Mycological tests in NICU patients.

	2005	2015	2016	2017	2018	2019	2020
Number of patients (with fungal cultures)	64	76	65	70	51	68	64
Female	29	42	29	40	21	37	28
Male	35	34	36	30	30	31	36
Mycological tests	133	133	115	124	104	113	119
Number of patients (with mycologic serological tests)	0	4	1	1	1	2	2
Female	0	2	0	1	1	2	1
Male	0	2	1	0	0	0	1
Serological tests	0	7	1	4	1	7	3

**Table 3 ijerph-19-01716-t003:** Yeast species isolated from the upper respiratory tract samples from PCICU each year.

Year (Amount of Tests)	2005 [6]	2015 [3]	2016 [7]	2017 [4]	2018 [1]	2019 [5]	2020 [1]
Species	Amount of Isolates (Percentage of Tests)						
*CANDIDA ALBICANS*	2 (33%)	2 (67%)	2 (29%)	2 (50%)	1 (100%)	2 (40%)	1 (100%)
*CANDIDA DUBLINIENSIS*	0	0	0	1	0	0	0
*CANDIDA KRUSEI*	0	0	0	1	0	0	0

**Table 4 ijerph-19-01716-t004:** Yeast species isolated from the upper respiratory tract samples from NICU patients.

Year (Amount of Tests)	2005 [1]	2015 [2]	2016 [7]	2017 [0]	2018 [1]	2019 [1]	2020 [3]
Species	Amount of Isolates (Percentage of Tests)						
*CANDIDA ALBICANS*	0	2 (100%)	1 (14%)	0	1 (100%)	0	0

**Table 5 ijerph-19-01716-t005:** Yeast species isolated from the lower respiratory tract samples in PCICU patients.

Year (Amount of Tests)	2005 [114]	2015 [135]	2016 [222]	2017 [313]	2018 [298]	2019 [157]	2020 [117]
Species	Amount of Isolates (Percentage of Tests)						
*CANDIDA ALBICANS*	17 (14.9%)	19 (14.1%)	19 (8.6%)	31 (9.9%)	21 (7%)	21 (13.4%)	11 (9.4%)
*CANDIDA DUBLINIENSIS*	0	5 (3.7%)	5 (2.3%)	7 (2.2%)	12 (4%)	8 (5%)	1 (0.9%)
*CANDIDA PARAPSILOSIS*	3 (2.6%)	2 (1.5%)	4 (1.8%)	12 (3.8%)	4 (1.3%)	3 (1.9%)	3 (2.6%)
*CANDIDA LUSITANIAE*	0	3 (2.2%)	5 (2.3%)	3 (1%)	5 (1.7%)	1 (0.6%)	2 (1.7%)
*CANDIDA GLABRATA*	0	3 (2.2%)	5 (2.3%)	3 (1%)	2 (0.7%)	1 (0.6%)	0
*CANDIDA KRUSEI*	0	0	0	2 (0.6%)	1 (0.3%)	3 (1.9%)	1 (0.9%)
*CANDIDA TROPICALIS*	0	1 (0.7%)	2 (0.9%)	2 (0.6%)	4 (1.3%)	0	2 (1.7%)

**Table 6 ijerph-19-01716-t006:** Yeast species isolated in NICU patients from the lower respiratory tract.

Year (Amount of Tests)	2005 [8]	2015 [17]	2016 [37]	2017 [47]	2018 [19]	2019 [31]	2020 [34]
Species	Amount of Isolates (Percentage of Tests)						
*CANDIDA ALBICANS*	0	1 (6%)	5 (14%)	3 (6%)	2 (11%)	5 (16%)	3 (9%)
*CANDIDA GUILLIERMONDII*	0	0	0	0	1 (5%)	0	4 (12%)
*CANDIDA GLABRATA*	0	0	0	0	0	1	1 (3%)
*CANDIDA KRUSEI*	0	0	0	1 (2%)	0	0	0
*CANDIDA PARAPSILOSIS*	0	0	1 (3%)	0	0	0	0

**Table 7 ijerph-19-01716-t007:** Yeast species isolated in PCICU patients from the urinary tract.

Year (Amount of Tests)	2005 [51]	2015 [95]	2016 [84]	2017 [86]	2018 [84]	2019 [75]	2020 [112]
Species	Amount of Isolates (Percentage of Tests)						
*CANDIDA ALBICANS*	9 (18%)	5 (5%)	7 (8%)	7 (8%)	12 (14%)	5 (7%)	6 (5%)
*CANDIDA PARAPSILOSIS*	1 (2%)	0	1 (1%)	7 (8%)	3 (4%)	1 (1%)	5 (4%)
*CANDIDA GLABRATA*	1 (2%)	8 (8%)	0	7 (8%)	0	2 (3%)	3 (3%)
*CANDIDA TROPICALIS*	0	9 (9%)	0	0	0	0	0
*CANDIDA MELIBIOSICA*	0	3 (3%)	0	0	0	0	0
*CANDIDA GUILLIERMONDII*	0	2 (2%)	0	0	0	1 (1%)	0
*CANDIDA KRUSEI*	0	1 (1%)	0	0	0	0	0
*CANDIDA DUBLINIENSIS*	0	1 (1%)	0	0	1 (1%)	1 (1%)	0
*CANDIDA LUSITANIAE*	0	0	0	0	0	0	4 (4%)

**Table 8 ijerph-19-01716-t008:** Yeast species isolated in NICU patients from the urinary tract.

Year (Amount of Tests)	2005 [67]	2015 [43]	2016 [25]	2017 [26]	2018 [42]	2019 [18]	2020 [17]
Species	Amount of Isolates (Percentage of Tests)						
*CANDIDA ALBICANS*	10 (15%)	9 (21%)	3 (12%)	2 (8%)	10 (24%)	0	1 (6%)
*CANDIDA PARAPSILOSIS*	2 (3%)	0	1 (4%)	0	0	0	0
*CANDIDA GLABRATA*	1 (1%)	0	0	0	0	0	0
*CANDIDA KRUSEI*	0	0	0	3 (12%)	0	0	0

**Table 9 ijerph-19-01716-t009:** Anti-Aspergillus antibodies (IgG) in PCICU patients.

Year	Positive Results	Number of Tested Patients	Percentage of Positive Results
2015	0	17	0
2016	0	13	0
2017	1	8	12.5%
2018	0	4	0
2019	0	3	0
2020	0	3	0

**Table 10 ijerph-19-01716-t010:** Galactomannan Aspergillus antigen in PCICU patients.

Year	Positive Results	Number of Tested Patients	Percentage of Positive Results
2015	1	44	2.27%
2016	0	35	0
2017	2	33	6.06%
2018	0	25	0
2019	0	37	0
2020	0	29	0

**Table 11 ijerph-19-01716-t011:** Candida mannan antigen in PCICU patients.

Year	Positive Results	Number of Tested Patients	Percentage of Positive Results
2015	9	70	12.86%
2016	3	57	5.26%
2017	1	46	2.17%
2018	2	35	5.71%
2019	7	50	14%
2020	0	34	0

**Table 12 ijerph-19-01716-t012:** Anti-mannan Candida antibodies in PCICU patients.

Year	Positive Results	Number of Tested Patients	Percentage of Positive Results
2015	0	22	0
2016	1	16	6.25%
2017	2	13	15.38%
2018	1	7	14.29%
2019	0	5	0
2020	0	4	0

**Table 13 ijerph-19-01716-t013:** Galactomannan Aspergillus antigen in NICU patients.

Year	Positive Results	Number of Tested Patients	Percentage of Positive Results
2015	0	1	0
2016	0	0	0
2017	0	0	0
2018	0	0	0
2019	0	2	0
2020	0	1	0

**Table 14 ijerph-19-01716-t014:** Anti-Aspergillus antibodies (IgG) in NICU patients.

Year	Positive Results	Number of Tested Patients	Percentage of Positive Results
2015	0	0	0
2016	0	0	0
2017	0	0	0
2018	0	0	0
2019	0	1	0
2020	0	0	0

**Table 15 ijerph-19-01716-t015:** Candida mannan antigen in NICU patients.

Year	Positive Results	Number of Tested Patients	Percentage of Positive Results
2015	2	3	66.67
2016	0	1	0
2017	0	1	0
2018	0	1	0
2019	0	1	0
2020	0	2	0

**Table 16 ijerph-19-01716-t016:** Anti-mannan Candida antibodies in NICU patients.

Year	Positive Results	Number of Tested Patients	Percentage of Positive Results
2015	0	0	0
2016	0	0	0
2017	0	0	0
2018	0	0	0
2019	0	1	0
2020	0	0	0

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
