# Peer review of "Fungal Infection Testing in Pediatric Intensive Care Units—A Single Center Experience"

_ijerph, 2022, doi:10.3390/ijerph19031716_

Round 1

Reviewer 1 Report

Dear Authors,

in my opinion your work is very interesting in a cognitive context and contributes a lot to mycology and epidemiology of invasive fungal infections (IFIs).

All the tables and figures are appropriate for this type of article. In general, the paper has a logical flow and it is refined in detail. The abstract well correspond with the main aspects of the work. Nevertheless, I see a few weak points in this work (given below), which I am convinced that the Authors are able to resolve very fast.

First of all, in the substantive context, I see two points, which would raise the value of this work.

  1. Introduction should be expanded, which will enrich the work with new references. I especially miss the references dealing with the prevalence of the etiological factors of fungal infections in the world in the group of patients studied by Authors. Also information on risk factors predisposing to IFIs in children would be appreciated.
  2. As for this type of work, too few publications by other authors are cited (only 11 references).

As a reviewer I am obligated to pay attention even to less important weak points of this work and all mentioned below comments should be carefully considered.

First of all, I want to point out that throughout the whole manuscript species names of fungi should be written in italics. Moreover, I would like to recommend to replace the term ,,fungal tests" on ,,mycological tests" throughout the whole work and in table headings. The species names of fungi used for the first time in the text should be written in their entirety without shortening the generic names, e.g. Candida glabrata.

Line 22

Something is missing here and in my opinion should be ,, ... were performed in the intensive care units "

Line 31

I would recommend replacing the keyword "mycological testing" with "mycological diagnostics"

Line 64

In my opinion correct is ,,…fungal infections”

Line 67

In my opinion ,,… compared with previous data collected in 2005” sounds better than ,,compared with historical data collected in 2005”

Line 72

I recommend leaving only "cultured fungi" because of all the yeasts are fungi and such wording is confusing

Line 96

In my opinion should be ,,…cultures in liquid and on solid Sabouraud medium”

Lines 93-117

In my opinion, it is not enough to just mention what "Mycological tests were performed" but at least briefly describe the techniques used, applied methods and quote according to which procedures.

Lines 138-139

Figure 4 is not clearly legible and should be improved in quality.

Line 167

The correct format is C. glabrata

Lines 169-170 and Table 5

According to the nomenclature, the correct names are Clavispora lusitaniae and Candida krusei.

Line 173 and Figure 9

Saccharomyces, but what species name? or maybe different species? Then should be Saccharomyces spp.

Lines 340-341

Based on their results Authors conclude that (quote) ,,Our experience indicates that the percentage of positive mycological tests in pediatric and neonatal intensive care units reaches 18-29%." On the pages of the discussion, it would be good to relate this information to literature data on this subject. How did it look in the research of other Authors?

Author Response

Dear Reviewer,

Thank you very much for your suggestions on how to improve our work. We have changed the lines that you recommended. We have also added a few publications. 

If you have any further recommendations please let us know. 

Reviewer 2 Report

This manuscript analyzed the epidemiology of fungal infection in the intensive care unit (Pediatric and Cardiac Surgery Intensive Care Unit-PCICU) and neonatal intensive care unit (NICU) of a large pediatric center from 2015 to 2020, and compares it with 2005. The results of this study have certain significance for the clinical diagnosis and treatment of fungal infections. But there are some shortcomings as follows:
1. The discussion part of the manuscript is relatively superficial and does not have sufficient statistical  analysis.
2.  The references cited in the article are not new enough, which is one of the reasons why the analysis is not in-depth.

Author Response

Dear Reviewer,

Thank you very much for your valuable suggestions. We have added some references to the discussion and improved writing style. If anything else should be improved please let us know.